# Impact of Palliative Gastrectomy in Patients with Incurable Gastric Cancer

**DOI:** 10.3390/medicina57030198

**Published:** 2021-02-26

**Authors:** Ji Yeon Park, Byunghyuk Yu, Ki Bum Park, Oh Kyoung Kwon, Seung Soo Lee, Ho Young Chung

**Affiliations:** 1Department of Surgery, School of Medicine, Kyungpook National University, Daegu 41944, Korea; jybark99@hanmail.net (J.Y.P.); parkeagle82@naver.com (K.B.P.); peterleess@hanmail.net (S.S.L.); hychung@mail.knu.ac.kr (H.Y.C.); 2Gastric Cancer Center, Kyungpook National University Chilgok Hospital, Daegu 41404, Korea; Johnyu421@gmail.com; 3Department of Surgery, Kyungpook National University Hospital, Daegu 41944, Korea

**Keywords:** stomach neoplasms, neoplasm metastasis, palliative care, gastrectomy

## Abstract

*Background and Objectives*: The prognosis of metastatic or unresectable gastric cancer is dismal, and the benefits of the palliative resection of primary tumors with noncurative intent remain controversial. This study aimed to evaluate the impact of palliative gastrectomy (PG) on overall survival in gastric cancer patients. *Materials and Methods*: One hundred forty-eight gastric cancer patients who underwent PG or a nonresection (NR) procedure between January 2011 and 2017 were retrospectively reviewed to select and analyze clinicopathological factors that affected prognosis. *Results*: Fifty-five patients underwent primary tumor resection with palliative intent, and 93 underwent NR procedures owing to the presence of metastatic or unresectable disease. The PG group was younger and more female dominant. In the PG group, R1 and R2 resection were performed in two patients (3.6%) and 53 patients (96.4%), respectively. The PG group had a significantly longer median overall survival than the NR group (28.4 vs. 7.7 months, *p* < 0.001). Multivariate analyses revealed that the overall survival was significantly better after palliative resection (hazard ratio (HR), 0.169; 95% confidence interval (CI), 0.088–0.324; *p* < 0.001) in patients with American Society of Anesthesiologists Physical Status (ASA) scores ≤1 (HR, 0.506; 95% CI, 0.291–0.878; *p* = 0.015) and those who received postoperative chemotherapy (HR, 0.487; 95% CI, 0.296–0.799; *p* = 0.004). Among the patients undergoing palliative resection, the presence of <15 positive lymph nodes was the only significant predictor of better overall survival (HR, 0.329; 95% CI, 0.121–0.895; *p* = 0.030). *Conclusions*: PG might lead to the prolonged survival of certain patients with incurable gastric cancer, particularly those with less-extensive lymph-node metastasis.

## 1. Introduction

Since the launch of the nationwide screening program in Korea, the detection of patients with far-advanced gastric cancer has gradually decreased over the last two decades, but consequently, there has been a rapid increase in the detection of patients with early gastric cancer. However, approximately 11% of all gastric cancer patients in Korea still suffer from advanced gastric cancer, which cannot be surgically resected with curative intent, and this corresponds to an annual estimate of >3000 newly diagnosed patients [1,2]. The prognosis of incurable gastric cancer is dismal, with a poor 5-year overall survival rate of <5% and a median survival of 10 months with palliative chemotherapy alone [3]. Many therapeutic modalities, including chemotherapy, radiation therapy, and immunotherapy, have been proposed to improve the prognosis of unresectable gastric cancer, but they showed limited effect in prolonging the survival of such patients. The optimal therapeutic strategies for unresectable or metastatic gastric cancer remain controversial.

The role of palliative resection in incurable gastric cancer has been debated over a prolonged period, and a consensus has not yet been reached. Based on previous studies that have demonstrated prolonged survival following surgical intervention, some investigators have advocated surgical resection in carefully selected stage IV patients [4,5]. However, most previous studies involved retrospective analyses and were subject to inevitable selection bias; patients with resectable tumors of less-extensive disease could have selectively received surgical resection after exploration. A sole randomized clinical trial (REGATTA) demonstrated that noncurative surgery led to no survival benefit in comparison with the survival associated with chemotherapy alone in gastric cancer patients with a single noncurative factor [6]. Nonetheless, efforts are ongoing toward the elucidation of the role of surgical intervention in incurable gastric cancer patients [7].

The present study aimed to both identify the prognostic factors for incurable gastric cancer and evaluate the impact of palliative gastrectomy (PG) on overall survival in incurable gastric cancer patients.

## 2. Materials and Methods

### 2.1. Patients

A total of 2985 consecutive patients who underwent surgery for preoperatively diagnosed gastric cancer at Kyungpook National University Chilgok Hospital, Daegu, Republic of Korea, between January 2011 and 2017 were retrospectively reviewed. All the patients underwent preoperative staging basically based on the findings from endoscopy and computed tomography of the abdomen and pelvis. When distant metastasis was suspected, additional positron emission tomography–computed tomography (PET–CT) scans were taken to facilitate more-precise staging and to avoid unnecessary surgical exploration in those with extensive tumor burdens such as extensive peritoneal seeding. Patients with evident distant metastasis were recommended to receive palliative chemotherapy first unless they showed any symptoms or signs warranting surgical management. 

We identified 148 patients who underwent surgery for incurable gastric cancer, as confirmed during surgical exploration; incurability was determined when the patient had distant metastasis or an unresectable primary tumor. Lymph-node metastasis beyond the boundaries of D2 lymphadenectomy, as recommended by the Japanese Gastric Cancer Treatment Guidelines, was also considered as an incurable disease [8].

These 148 patients with incurable disease were classified into two groups, namely, the PG group and nonresection (NR) group, according to the surgical procedure provided. The decision on which procedure should be administered was made with the consent of the patients’ guardians after providing them with a sufficient explanation of the patients’ disease statuses and prognoses. 

Clinicopathological information and surgical outcomes were retrieved by performing a thorough retrospective review of the medical records of the included patients. 

The patients were followed up every 3 months in the first year after surgery, every 6 months up to 3 years, and annually thereafter. Laboratory tests including those of tumor markers were routinely performed at each visit, and abdominopelvic CT scans were regularly taken every 6 months to evaluate the disease status. Patient survival was determined by reviewing the medical records from follow-up visits or by contacting the patients over the telephone. Overall survival was recorded as the time from surgery to death regardless of the cause or until the time of the last follow-up visit. The patients were followed up until death or October 1st, 2017. The median follow-up period was 10.9 months (range, 0.2–69.5 months). 

Approval for this review of medical records was obtained from the institutional review board (KNUCH 2019-08-030, approved on 5 September 2019), and the need for patient informed consent was waived. 

### 2.2. Statistical Analyses

All statistical analyses were conducted using the SPSS^®^ software (version 18.0; IBM, Chicago, IL, USA). Patients in both the PG and NR groups were matched using propensity score matching (PSM) to reduce the possibility of selection bias for comparisons. The logistic regression model was used to calculate the propensity scores for each patient based on the baseline characteristics including age, sex, American Society of Anesthesiologists Physical Status (ASA-PS) score, histologic differentiation, and reasons for incurability. Patients in the PG and NR groups were matched at a 1:1 ratio using the nearest propensity scores on the logit scale. Either Pearson’s chi-square test or Fisher’s exact test was used to compare categorical variables, whereas Student’s *t*-test or the Mann–Whitney U test was applied to evaluate the differences in continuous variables between the groups. The Kaplan–Meier method was used to estimate the overall survival in each group, and the difference was assessed using the log-rank test. The Cox regression model was used for conducting univariate and multivariate analyses to identify independent prognostic factors for overall survival. Two-sided *p*-values were calculated for all the tests, and a *p*-value < 0.05 was considered to be statistically significant.

## 3. Results

With regard to the baseline characteristics, the mean age of all 148 patients was 60.6 ± 14.1 years, and 64.2% of them were men (Table 1). A majority of the patients (91.9%) had favorable physical statuses according to their ASA-PS scores. At the time of initial disease assessment, 14 patients (9.5%) presented with gastric outlet obstruction, and five patients (3.4%) had cancer bleeding. Preoperative radiologic evaluation via abdominal computed tomography scans and/or PET-CT scans revealed that 19 patients (12.8%) had far-advanced primary tumors directly invading adjacent organs, whereas 33 patients (22.3%) had suspicious metastatic disease. Ten patients (6.8%) received preoperative chemotherapy for the initially diagnosed stage IV disease and were then referred for surgery for various reasons: a good response to chemotherapy (*n* = 5), intractable bleeding (*n* = 1), outlet obstruction (*n* = 2), perforation (*n* = 1), and poorly controlled pain (*n* = 1). 

Among them, 55 patients underwent palliative resection for incurable gastric cancer (PG group) during the study period, whereas 93 patients received nonresectional procedures (NR group), including palliative gastrojejunostomy and simple exploration (Table 1). Twenty-four patients (16.2%) underwent extended surgery involving the resection of other organs. In most patients (51/55, 92.7%) undergoing palliative resection, lymph-node dissections of D1+ or more were carried out to achieve at least R1 resection. The most predominant cause of incurability was peritoneal seeding, accounting for 67.6% of the patients, followed by advanced primary tumor invasion into adjacent organs (43.9%). Forty-eight (32.4%) of the patients had two or more noncurative factors. 

A comparison between the PG and NR groups revealed that the patients in the PG group were significantly younger than those in the NR group by approximately 10 years (54.0 vs. 64.5 years; *p* < 0.001) (Table 2). However, patients with poorer ASA scores, ≥3, were more frequently encountered in the PG group (14.5% vs. 4.3%; *p* < 0.041). The NR group had a significantly larger number of patients with unresectable tumors owing to direct infiltration into adjacent organs (54.1%) compared with the PG group (34.5%, *p* = 0.025); the incidence of distant metastasis, including distant lymph nodes and other organs, was similar between the groups. Although overall postoperative complications were more prevalent in the PG group (20.0% vs. 3.2%, *p* < 0.001), most of them were well managed with conservative measures. A severe complication of grade III or more occurred in only one patient (0.7%), who required relaparotomy to control immediate postoperative bleeding on the first postoperative day after undergoing a palliative Whipple’s procedure. Most of the patients in the PG group (85.5%) received postoperative chemotherapy after the palliative surgery, whereas significantly fewer patients (59.1%) received chemotherapy after nonresectional procedures (*p* = 0.017). Among the patients who underwent at least one cycle of postoperative chemotherapy, the patients in the PG group completed a significantly larger number of chemotherapy cycles than those in the NR group (median, 13 vs. 8 cycles, *p* = 0.004). 

The PSM analyses included 55 patients in each group. A comparison between the two groups after propensity score matching demonstrated that the two groups were well balanced in terms of preoperative disease status (Table 2). However, the patients in the PG group were still younger than the NR group after matching (54.0 vs. 59.9 years, *p* = 0.020). 

The median survival of the whole study cohort after propensity score matching was 14.4 months (95% confidence interval (CI), 11.5–17.3 months). Table 3 presents the prognostic factors for the overall survival after matching. Univariate analysis revealed that the better survival of these patients with incurable gastric cancer was associated with the need for undergoing palliative resection, an age less than 70 years, an ASA physical status score ≤1, and the administration of postoperative chemotherapy (Table 3). Multivariate analyses conducted after adjusting for confounding factors revealed that palliative resection (hazard ratio (HR) = 0.169; 95% CI, 0.089–0.321) and a preoperative ASA score ≤1 (HR = 0.425; 95% CI, 0.215–0.838) were independently correlated with an improved overall survival. Noticeably, age < 70 years and postoperative chemotherapy were not proven as independent prognostic factors in the multivariate analysis. The median overall survival periods of the PG and NR groups after propensity score matching were 28.4 months (95% CI, 10.9–45.9 months) and 7.6 months (95% CI, 6.6–8.6 months), respectively, with the former group showing significantly better survival after palliative resection (log-rank test, *p* < 0.001) (Figure 1). 

Univariate analysis revealed that the presence of < 15 metastatic lymph nodes and extended nodal disease beyond second-tier lymph-node stations appeared to be significantly related to better overall survival in the patients who underwent palliative resection, while a younger age (age < 70 years), good physical status (ASA ≤ 1), and administration of postoperative chemotherapy showed a marginal significance (Table 4). Multivariate analysis revealed that the presence of <15 metastatic lymph nodes (HR = 0.329; 95% CI, 0.121–0.895) and administration of postoperative chemotherapy (HR, 0.258; 95% CI, 0.068–0.984) were independent prognostic factors for overall survival after palliative resection. The median overall survival was 19.2 months (95% CI, 12.7–25.7 months) in patients with ≥15 metastatic lymph nodes following PG, which was significantly shorter than that of patients with lesser numbers of metastatic lymph nodes (*p* = 0.013, log-rank test) (Figure 2A). Forty-seven patients (85.5%) received chemotherapy after palliative gastrectomy, and their median survival was 38.3 months (95% CI, 18.9–57.7 months), which was significantly longer than the 11.7 months (95% CI, 0.0–31.5 months) of those who did not receive postoperative chemotherapy (*p* = 0.044, long-rank test) (Figure 2B).

## 4. Discussion

The present study demonstrated that PG could lead to prolonged survival in patients with incurable gastric cancer. Although R0 resection could not be achieved owing to the extent of disease at baseline, reducing the tumor burden with palliative intent appeared to result in superior survival as opposed to the administration of palliative chemotherapy alone or providing the best supportive care.

When incurability is confirmed during surgical exploration, it is difficult to decide whether or not to perform palliative resection. According to a previous systematic review and meta-analyses, patients with advanced gastric cancer who underwent surgical resection had better survival than those who were treated conservatively, and this finding is consistent with the results of the present study [4,5]. Although most of the studies included in previously published reviews were retrospective studies with relatively small sample sizes and could have been heavily biased in favor of surgical resection, the results of those studies imply that a deliberately selected subset of patients with incurable gastric cancer might benefit from the surgical resection of the primary tumor with or without extensive lymph-node dissection.

Contrastingly, a previously conducted randomized clinical trial, the REGATTA trial, discouraged noncurative surgery in patients with stage IV gastric cancer with a single incurable factor because noncurative surgery followed by chemotherapy provided no survival benefit in comparison with chemotherapy alone in these patients [6]. However, it should be noted that the patients included in this clinical trial were carefully selected from those with a burden of primary tumors at lower stages up to T3; patients with far-advanced primary tumors showing serosal exposure or any evidence of adjacent organ invasion were not included in this trial. Patients with cancer-related complications, including bleeding and obstruction, which result in poor oral intake, were also excluded from this trial. Therefore, in this trial, the patients in the chemotherapy-alone group were expected to tolerate chemotherapy sufficiently well without palliative resection and be less likely to experience cancer-related symptoms that frequently result in the interruption of chemotherapy. Additionally, the patients in the combined surgery and chemotherapy group underwent surgical intervention to a lesser extent along with D1 lymph-node dissection while the metastatic lesions were left untouched. This could have had a negative influence on the overall survival of the patients who underwent noncurative surgery.

The impact of surgical resection on compliance with postoperative chemotherapy remains controversial. Some previous studies have demonstrated that surgical resection resulted in poor compliance with postoperative chemotherapy owing to poor performance status and poor oral intake followed by gastrectomy [6,9]. Bodyweight loss induced by gastrectomy was reported to be a risk factor for the interruption of postoperative chemotherapy [10]. Extensive surgery can cause postoperative complications and prolonged recovery that subsequently leads to a delay in systemic chemotherapy [11]. However, some investigators have put forth a contradictory opinion that surgical resection helps these patients to avoid cancer-related symptoms or complications such as outlet/inlet obstruction or cancer bleeding, owing to which the patients are less likely to encounter the interruption of postoperative palliative chemotherapy [12]. A similar effect was shown in the present study cohort; the patients in the PG group were more likely to comply with the scheduled chemotherapy and completed a significantly higher number of chemotherapy cycles, which could have resulted in survival outcomes better than those of the patients in the NR group. The compliance with postoperative chemotherapy appears to have been dependent on the palliative resection and the better physical status. The patients in the NR group were not only less likely to start palliative chemotherapy but also less likely to tolerate the treatment in the long run. These patients with unresected primary cancer experienced extensive disease progression at early stages even during palliative chemotherapy. Therefore, a carefully selected subset of patients with favorable performance statuses might benefit from surgical tumor resection, whenever technically feasible, by avoiding debilitating cancer-related symptoms and by prolonging the time to disease progression during the course of palliative chemotherapy. Consequently, palliative resection might provide a better quality of life, although a statistically significant increase in the survival period may not be achieved. A previous study involving 150 gastric cancer patients undergoing palliative procedures demonstrated that patients who underwent resection had better palliation of the symptoms and a significantly better quality of life than those undergoing nonresectional procedures [13].

Some investigators have suggested that preoperative chemotherapy followed by curative surgery improved survival in patients with incurable gastric cancer. The concept adopted here can be referred to as conversion surgery, which can be characterized as surgical treatment with curative intent administered after chemotherapy for tumors that were initially deemed either oncologically or technically unresectable. In our study cohort, resection was possible only in 4 of 10 patients who underwent surgery after the administration of upfront chemotherapy for incurable gastric cancer. Two of those patients could be referred for conversion surgery, but the other two patients required unscheduled palliative resection owing to cancer bleeding and outlet obstruction. However, the number of patients who received preoperative chemotherapy was extremely small in the present study, which means patients with metastatic gastric cancer were rarely referred for surgical resection during palliative chemotherapy. The decisions to attempt surgical resection after upfront chemotherapy were neither consistent nor based on the responses to systemic chemotherapy, and, therefore, it is difficult to estimate the efficacy of systemic chemotherapy followed by palliative resection based on the present study. This indirectly reflects that the indications for conversion surgery have not yet been clarified, and at present, there is limited clinical experience to support the efficacy of conversion therapy in a single institution alone. Nonetheless, previous studies have reported that approximately 20–30% of patients with incurable gastric cancer were eligible for conversion surgery after upfront systemic chemotherapy and that R0 resection appeared to be correlated with improved survival, with the median survival period reaching up to 60 months [12,14,15,16,17,18]. The results of recently published Korean retrospective studies are consistent with the abovementioned findings, suggesting that the survival benefit of conversion surgery after chemotherapy was particularly obvious when R0 resection was achieved in chemoresponsive patients [19,20]. These results indicate that intimate collaboration is essential in a multidisciplinary team for selecting suitable patients for conversion surgery with curative intent after upfront chemotherapy.

Because stage IV gastric cancer is prevalent in considerably heterogeneous subgroups of patients, it is questionable whether patients with stage IV gastric cancer demonstrate similar prognoses regardless of the mode or extent of distant metastasis. Yoshida et al. have proposed a comprehensive classification of stage IV gastric cancer based on possible therapeutic strategies, and it is believed to be able to facilitate future investigations for clarifying the role of surgery in stage IV gastric cancer by reducing the risk of critical selection bias [21].

There are several limitations to the present study. First, the retrospectively collected data might have resulted in a selection bias. The patients in the PG group might have been more likely to have lower disease burdens and technically resectable tumors than those in the NR group, among whom tumor resection was mostly impossible or futile in the first place. Furthermore, the patients in the PG group could have had better performance statuses than those in the NR group, and they were also sufficiently healthy for tolerating extensive surgery as well as chemotherapy, which can be inferred indirectly from the difference in baseline age between the two groups; therefore, the former group had a better prognosis even before treatment initiation. This selection bias could have led to an overestimation of the palliative resection benefits in the present study. Although the propensity-score matching analysis method was adopted to reduce the selection bias, the results from the present study should be interpreted with great caution. Second, the present study included patients who underwent truly palliative surgery with an intention of symptom relief as well as those who underwent cytoreductive surgery with curative intent after induction chemotherapy. Thus, it was noteworthy that there was a wide variation in disease characteristics among these patients at the time of surgery. Although it was impossible to conduct stratified analyses in the present study owing to the small sample size, relevant prospective studies in the future could attain a definite conclusion on this issue.

## 5. Conclusions

The present study suggested that PG facilitated the prolonged survival of carefully selected patients with incurable gastric cancer, and patients with less-extensive nodal metastasis demonstrated significant survival benefits following palliative resection. Although several issues pertaining to the treatment of incurable gastric cancer and the role of palliative surgery are yet to be clarified, the careful assessment of individual patients by a multidisciplinary team, including a surgeon, oncologist, and radiologist, might accelerate the development of a more-effective therapeutic strategy and eventually achieve survival improvement in patients with this dismal disease.

## Figures and Tables

**Figure 1 medicina-57-00198-f001:**
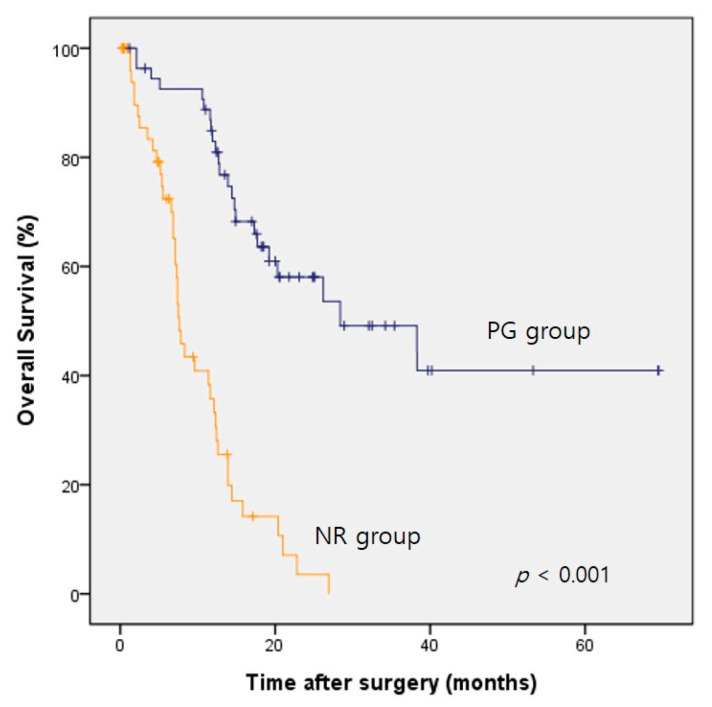
Comparison of overall survival in patients undergoing palliative resection vs. nonresection after propensity score matching. PG, palliative gastrectomy; NR, nonresection (*p* < 0.001, log-rank test).

**Figure 2 medicina-57-00198-f002:**
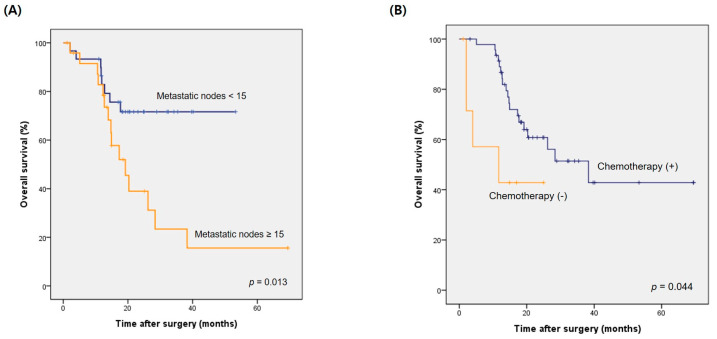
Difference in overall survival in patients who underwent palliative resection: (**A**) according to the number of metastatic lymph nodes (*p* = 0.013); (**B**) according to the administration of postoperative chemotherapy (*p* = 0.044, log-rank test).

**Table 1 medicina-57-00198-t001:** The clinical characteristics of the patients and their surgical findings (*n* = 148).

Variables	No. of Patients (%)
Age (years)	60.6 ± 14.1
**Sex**	
Male	95 (64.2)
Female	53 (35.8)
Body mass index (kg/m^2^)	22.2 ± 3.2
**ASA-PS score**	
1	51 (34.5)
2	85 (57.4)
3	11 (7.4)
4	1 (0.7)
**Histology**	
Differentiated	39 (26.4)
Undifferentiated	106 (71.6)
Unknown	3 (2.0)
**Gross type**	
Bormann type I	1 (0.7)
Bormann type II	13 (8.8)
Bormann type III	77 (52.0)
Bormann type IV	38 (25.7)
Unspecified	19 (12.8)
Metachronous cancer in the remnant stomach	6 (4.1)
Outlet obstruction	14 (9.5)
Bleeding from cancer	5 (3.4)
**Preoperative radiologic findings**	
Adjacent organ invasion	19 (12.8)
Metastatic disease	33 (22.3)
**Tumor markers at baseline**	
CEA (ng/mL)	1.8 (0.4–255.3)
CA 19-9 (U/mL)	15.9 (0.2–1822.9)
CA 125 (U/mL)	8.8 (0.9–200.8)
**Type of surgery**	
Palliative resection	55 (37.2)
Distal gastrectomy	28 (18.9)
Total gastrectomy	25 (16.9)
Whipple’s operation	2 (1.4)
Nonresection	93 (62.8)
Bypass surgery (including gastrojejunostomy)	39 (26.4)
Opening and closure	54 (36.5)
Combined resection	24 (16.2)
**Extent of lymph-node dissection**	
D0	93 (62.8)
Less than D2	4 (2.7)
D2 or more	51 (34.5)
**Reason for incurability**	
Advanced primary disease	65 (43.9)
Extended nodal disease	21 (14.2)
Metastatic disease	120 (81.1)
Peritoneal seeding	100 (67.6)
Hepatic metastasis	9 (6.1)
Metastasis to other organs (except for liver)	9 (6.1)
Two or more noncurative factors	48 (32.4)
Intraperitoneal chemotherapy †	14 (9.5)
Postoperative chemotherapy	102 (68.9)
Overall survival (months)	10.9 (0.2–69.5)

ASA-PS, American Society of Anesthesiologists Physical Status; CEA, carcinoembryonic antigen. † Intraperitoneal chemotherapy was carried out during the immediate postoperative period for 5 days via catheter installed intraoperatively.

**Table 2 medicina-57-00198-t002:** Comparison of the surgical outcomes between the groups before and after propensity score matching.

Variables	Before Matching			After Matching		
	PG Group (*n* = 55)	NR Group (*n* = 93)	*p*-Value	PG Group (*n* = 55)	NR Group (*n* = 55)	*p*-Value
Age at operation (years)	54.0 ± 14.1	64.5 ± 12.6	<0.001	54.0 ± 14.1	59.9 ± 12.1	0.020
**Sex**						
Male	28 (50.9%)	67 (72.0%)	0.013	28 (50.9%)	36 (65.5%)	0.122
Female	27 (49.1%)	26 (28.0%)		27 (49.1%)	19 (34.5%)	
**ASA-PS classification**			0.054			0.076
1	21 (38.2%)	30 (32.3%)		21 (38.2%)	16 (29.1%)	
2	26 (47.3%)	59 (63.4%)		26 (47.3%)	37 (67.3%)	
3	7 (12.7%)	4 (4.3%)		8 (14.5%)	2 (3.6%)	
4	1 (1.8%)	0 (0%)		1 (1.8%)	0 (0%)	
**Clinical stage ***						
T category			0.109			0.495
T2	3 (5.5%)	2 (2.2%)		3 (5.5%)	1 (1.8%)	
T3	9 (16.4%)	18 (19.4%)		9 (16.4%)	12 (21.8%)	
T4a	24 (43.6%)	26 (28.0%)		24 (43.6%)	19 (34.5%)	
T4b	19 (34.5%)	47 (50.5%)		19 (34.5%)	23 (41.8%)	
**N category**						0.311
N0	5 (9.1%)	8 (8.6%)	0.086	5 (9.1%)	4 (7.3%)	
N1	7 (12.7%)	24 (25.8%)		7 (12.7%)	13 (23.6%)	
N2	24 (43.6%)	45 (48.4%)		24 (43.6%)	27 (49.1%)	
N3a	18 (32.7%)	15 (16.1%)		18 (32.7%)	11 (20.0%)	
N3b	1 (1.8%)	1 (1.1%)		1 (1.8%)	0 (0%)	
**Reason for incurability**						
Advanced primary disease	19 (34.5%)	46 (54.1%)	0.025	19 (34.5%)	23 (41.8%)	0.432
Extended nodal disease	7 (12.7%)	10 (10.8%)	0.792	7 (12.7%)	7 (12.7%)	>0.999
**Metastatic disease**						
Peritoneal seeding	41 (74.5%)	59 (63.4%)	0.204	41 (74.5%)	37 (67.3%)	0.401
Hepatic metastasis	1 (1.8%)	8 (8.6%)	0.154	1 (1.8%)	1 (1.8%)	>0.999
Metastasis to other organs (except for liver)	5 (9.1%)	4 (4.3%)	0.293	5 (9.1%)	3 (5.5%)	0.716
**Two or more noncurative factors**	20 (36.4%)	28 (30.1%)	0.470	20 (36.4%)	13 (23.6%)	0.146
**Postoperative complications †**			<0.001			0.003
None	44 (80.0%)	90 (96.8%)		44 (80.0%)	54 (98.2%)	
Grade I	7 (12.7%)	1 (1.1%)		7 (12.7%)	0 (0%)	
Grade II	3 (5.5%)	2 (2.2%)		3 (5.5%)	1 (1.8%)	
Grade IV	1 (1.8%)	0		1 (1.8%)	0 (0%)	
Length of hospital stay (days)	13.0 ± 11.1	9.3 ± 7.3	0.017	13.0 ± 11.1	9.5 ± 8.0	0.063
Preoperative chemotherapy	4 (7.3%)	6 (6.5%)	>0.999	4 (7.3%)	5 (9.1%)	>0.999
Intraperitoneal chemotherapy	14 (25.5%)	0	<0.001	14 (25.5%)	0 (0%)	<0.001
Postoperative chemotherapy	47 (85.5%)	55 (59.1%)	0.001	47 (85.5%)	31 (56.4%)	0.001
No. of chemotherapy cycles ‡	13 (1–104)	8 (1–44)	0.004	13 (1–104)	8 (1–39)	0.027

PG, palliative gastrectomy; NR, nonresection; ASA-PS, American Society of Anesthesiologists Physical Status. Data are presented as *n* (%), mean ± SD, or median (range), as appropriate. * Clinical and pathologic stages were described according to the AJCC 7th edition. † Severity of postoperative complications was assessed according to the modified Clavien-Dindo grading system. ‡ Among those who received at least 1 cycle of chemotherapy.

**Table 3 medicina-57-00198-t003:** Univariate and multivariate analysis for the prognostic factors of overall survival in the propensity score-matched cohort with incurable gastric cancer.

Variables	Univariate Analysis		Multivariate Analysis	
	HR (95% CI)	*p*-Value	HR (95% CI)	*p*-Value
Palliative resection (vs. nonresection)	0.185 (0.106–0.321)	<0.001	0.169 (0.089–0.321)	<0.001
Age < 70 years (vs. age ≥ 70)	0.545 (0.307–0.969)	0.039	0.617 (0.322–1.181)	0.145
Male (vs. female)	1.382 (0.832–2.295)	0.211		
ASA-PS ≤ 1	0.476 (0.270–0.842)	0.011	0.425 (0.215–0.838)	0.014
Outlet obstruction	1.466 (0.631–3.409)	0.374		
Differentiated (vs. undifferentiated)	0.943 (0.527–1.685)	0.842		
**Tumor markers**				
CEA > 7	1.390 (0.502–3.849)	0.527		
CA 19-9 > 37	0.858 (0.476–1.545)	0.609		
CA 125 > 30	2.020 (0.908–4.494)	0.085	1.373 (0.532–3.543)	0.512
**Surgical findings**				
Advanced primary disease	1.127 (0.676–1.880)	0.646		
Extended nodal disease	1.435 (0.705–2.921)	0.319		
Peritoneal seeding	0.662 (0.389–1.127)	0.129		
Hepatic metastasis	0.627 (0.087–4.549)	0.645		
Neoadjuvant chemotherapy	1.983 (0.899–4.371)	0.090	0.797 (0.270–2.350)	0.681
Intraperitoneal chemotherapy	0.561 (0.255–1.234)	0.151		
Postoperative chemotherapy	0.326 (0.189–0.563)	<0.001	0.665 (0.360–1.231)	0.194

ASA-PS, American Society of Anesthesiologists Physical Status; CEA, carcinoembryonic antigen.

**Table 4 medicina-57-00198-t004:** Prognostic factors after palliative resection.

Variables	Univariate Analysis		Multivariate Analysis	
	HR (95% CI)	*p*-Value	HR (95% CI)	*p*-Value
Age < 70 years (vs. age ≥ 70)	0.432 (0.174–1.072)	0.070	1.170 (0.371–3.689)	0.788
Male (vs. female)	1.246 (0.545–2.848)	0.601		
ASA-PS ≤ 1	0.409 (0.152–1.104)	0.078	0.528 (0.181–1.537)	0.241
Differentiated (vs. undifferentiated)	0.850 (0.252–2.867)	0.793		
**Extent of surgery**				
Distal gastrectomy	Ref.	0.624		
Total gastrectomy	0.845 (0.365–1.956)	0.695		
Whipples’ operation	2.308 (0.292–18.252)	0.428		
LND ≥ D2 (vs. LND < D2)	1.527 (0.204–11.419)	0.680		
**Pathologic findings**				
No. of positive nodes < 15	0.351 (0.148–0.832)	0.014	0.327 (0.129–0.831)	0.019
Peritoneal seeding	0.862 (0.339–2.190)	0.755		
Extended nodal disease	3.472 (1.237–9.745)	0.018	3.078 (0.931–10.176)	0.065
Advanced primary disease	1.017 (0.415–2.495)	0.971		
Intraperitoneal chemotherapy	1.380 (0.559–3.407)	0.484		
Postoperative chemotherapy	0.339 (0.112–1.024)	0.055	0.258 (0.068–0.984)	0.047

ASA-PS, American Society of Anesthesiologists Physical Status; LND, lymph-node dissection.

## Data Availability

The data presented in this study are available on request from the corresponding author.

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
