# Peer review of "Impact of Palliative Gastrectomy in Patients with Incurable Gastric Cancer"

_medicina, 2021, doi:10.3390/medicina57030198_

Round 1

Reviewer 1 Report

I recently received your article for peer review evaluation: although I appreciated your paper and I think that the results that you present in this work are interesting, I also think that a few matters should be addressed before your work can be accepted for publication:

Major Issues:

Methodology is explained in detail and in a easily understandable way. As a matter of fact, since it is expected that metastatic gastric cancer patients receive some benefit from palliative chemotherapy, even though you correctly performed multivariate analysis to assess the independent role of both "palliative gastrectomy" and palliative chemotherapy, this is not sufficient. To correctly define the impact of palliative gastrectomy in patients who received chemotherapy, some form of matching analysis is necessary (I suggest propensity score matching). By using this method you would be able to correctly estimate the beneficial impact of palliative gastrectomy, as you would have a group of patients that "before surgery" would have had the same likelihood to receive also palliative chemotherapy. 

Minor Issues:

I perfectly agree with your comment about the limitations of REGATTA trial. Indeed, a few retrospective studies have suggested that for some selected few patients might benefit from this palliative surgery (and you cited a few of them). Although you have only a few patients that received pre-operative chemotherapy (10/148) and they were evenly distributed between those who received palliative gastrectomy and not-resectional surgical procedures (4 vs 6, p>0.999), this is somewhat surprising. As you stated in your paper, response to chemotherapy is one of the factors that might influence the decision to candidate a patient for palliative gastrectomy. This has been explained in some papers (such as Fornaro L, et al. Selecting patients for gastrectomy in metastatic esophago-gastric cancer: clinics and pathology are not enough. Future Oncol. 2017 Oct;13(25):2265-2275. doi: 10.2217/fon-2017-0246. Epub 2017 Oct 4). I would like to see some comment about the different results of your work compared with others were the choice to perform palliative gastrectomy is performed "after" that response to chemotherapy is achieved.

Furthermore, it is very interesting your finding that, in patients who underwent palliative gastrectomy, the only factor that was significantly associated with different survival outcomes was the number of positive lymphnodes. This is particularly interesting, since your population is comprised mainly of patients with peritoneal metastases and you would expect that the majority of these patients would die because of complications due to peritoneal seeding (more than because of other factors). I would like to know whether in patients with greater lymphnodal involvement worse survival outcomes might be due to different "pattern" of metastatic spread after surgery (as in occurrence of new metastatic lesions). 

Author Response

I recently received your article for peer review evaluation: although I appreciated your paper and I think that the results that you present in this work are interesting, I also think that a few matters should be addressed before your work can be accepted for publication:

Major Issues:

Methodology is explained in detail and in a easily understandable way. As a matter of fact, since it is expected that metastatic gastric cancer patients receive some benefit from palliative chemotherapy, even though you correctly performed multivariate analysis to assess the independent role of both "palliative gastrectomy" and palliative chemotherapy, this is not sufficient. To correctly define the impact of palliative gastrectomy in patients who received chemotherapy, some form of matching analysis is necessary (I suggest propensity score matching). By using this method you would be able to correctly estimate the beneficial impact of palliative gastrectomy, as you would have a group of patients that "before surgery" would have had the same likelihood to receive also palliative chemotherapy. 

  • As the reviewer has recommended, we conducted additional propensity score matching analysis to reduce the possibility of selection bias. The propensity score was calculated in each patient based on the preoperative factors, but not including postoperative variables as described in the revised manuscript. As a result, PG and NR groups were well-balanced in terms of disease extent. After matching, the palliative gastrectomy remained as the independent prognostic factor, however, postoperative chemotherapy did not. As described in the discussion section, administration of the postoperative chemotherapy appears to be dependent on the palliative resection and the better physical status.

Minor Issues:

I perfectly agree with your comment about the limitations of REGATTA trial. Indeed, a few retrospective studies have suggested that for some selected few patients might benefit from this palliative surgery (and you cited a few of them). Although you have only a few patients that received pre-operative chemotherapy (10/148) and they were evenly distributed between those who received palliative gastrectomy and not-resectional surgical procedures (4 vs 6, p>0.999), this is somewhat surprising. As you stated in your paper, response to chemotherapy is one of the factors that might influence the decision to candidate a patient for palliative gastrectomy. This has been explained in some papers (such as Fornaro L, et al. Selecting patients for gastrectomy in metastatic esophago-gastric cancer: clinics and pathology are not enough. Future Oncol. 2017 Oct;13(25):2265-2275. doi: 10.2217/fon-2017-0246. Epub 2017 Oct 4). I would like to see some comment about the different results of your work compared with others were the choice to perform palliative gastrectomy is performed "after" that response to chemotherapy is achieved.

  • As the reviewer has pointed out, only a few patients were referred to surgery after palliative chemotherapy, which means patients with metastatic gastric cancer rarely referred for surgical counseling during palliative chemotherapy courses unless they develop any cancer-related complications. Therefore, it is difficult to discuss the efficacy of overall upfront chemotherapy followed by palliative resection and additional comments were made in the discussion section.

Furthermore, it is very interesting your finding that, in patients who underwent palliative gastrectomy, the only factor that was significantly associated with different survival outcomes was the number of positive lymphnodes. This is particularly interesting, since your population is comprised mainly of patients with peritoneal metastases and you would expect that the majority of these patients would die because of complications due to peritoneal seeding (more than because of other factors). I would like to know whether in patients with greater lymph nodal involvement worse survival outcomes might be due to different "pattern" of metastatic spread after surgery (as in occurrence of new metastatic lesions). 

  • The majority of our patients (74.5%) in the PG group already had concurrent peritoneal metastasis at the time of surgery as shown in the Table 2. Intriguingly, the prevalence of peritoneal metastasis was not significantly different between the patients with smaller metastatic nodes (< 15) vs. those with bulky metastatic nodal burden (No. of metastatic lymph nodes >15) (73.3% vs. 75%). And as the reviewer has pointed out, almost all patients died of complications from peritoneal seeding which aggravated within several months after surgery (median 9.9 months), but rarely from other new metastatic lesions. It appears that the larger number of metastatic lymph nodes indicates a larger disease burden as well as a poorer response to palliative chemotherapy, and these seem to be associated with worse survival.

Reviewer 2 Report

In this manuscript Park et al investigated the effectiveness of palliative resection for incurable gastric cancers. They found that the patients who underwent palliative resection had better survival compared to the patients who did not.

1) As the authors stated in Discussion, the finding of this study could be largely dependent on the selection bias. In addition, unlike REGATTA trial, this study is a single centered study. For these reasons, we have to be cautious in interpreting the results of this study, and these limitations should be emphasized in the text.

2) Since the survival benefit of the patients in PG group could be due to the effect of chemotherapy, the authors should show the overall survival curve of post-operative chemotherapy in PG group and NR group.

Author Response

In this manuscript Park et al investigated the effectiveness of palliative resection for incurable gastric cancers. They found that the patients who underwent palliative resection had better survival compared to the patients who did not.

1) As the authors stated in Discussion, the finding of this study could be largely dependent on the selection bias. In addition, unlike REGATTA trial, this study is a single centered study. For these reasons, we have to be cautious in interpreting the results of this study, and these limitations should be emphasized in the text.

We totally agree with the reviewer’s opinion. The present study has a critical limitation as described in the discussion section; it is a single-centered, retrospectively study with critical selection bias. Further emphasis was made in the limitation at the end of the discussion section.

2) Since the survival benefit of the patients in PG group could be due to the effect of chemotherapy, the authors should show the overall survival curve of post-operative chemotherapy in PG group and NR group.

The survival curves in PG vs. NR groups including only those who received postoperative chemotherapy are presented below; the PG group (median OS 38.3 months, 95% CI 18.9 – 57.7 months) demonstrated still superior overall survival than the NR group (median OS 11.4 months, 95% CI 4.8 – 18.1 months). As described in the discussion section, administration of postoperative chemotherapy appears to be largely dependent on the palliative resection and the better physical status in this study population. After resection, however, postoperative chemotherapy significantly influenced on the overall survival as shown in the figure 2(B) in the revised manuscript.

Round 2

Reviewer 2 Report

The authors addressed the concerns.